materials science

humidity, permeability, $T_2$ spectral area, UCS, weighted average $T_2$

**Author for correspondence:**
Qifan Ren
e-mail: qifanren@csu.edu.cn

# Microscopic characterization and strength characteristics of cemented backfill under different humidity curing conditions

Jianhua HU[1], Fengwen Zhao[2], Qifan Ren[2], Ye Kuang[2], Tan Zhou[2] and Zhouquan Luo[1]

[1]School of Resources and Safety Engineering, and [2]Research Assistant, School of Resources and Safety Engineering, Central South University, Changsha, Hunan 410081, People's Republic of China

JH, 0000-0003-1293-6829; FZ, 0000-0001-9744-8888; QR, 0000-0002-9814-834X; TZ, 0000-0002-0792-1676

Under different exploitive conditions, the humidity levels of the backfill stopes are not the same. Humidity greatly affects the strength and microscopic characterization of the backfill. Cemented paste backfill (CPB) specimens were cured using 0, 30, 70% and standard curing (20°C, 99%) under four different humidity conditions. At 28 days, nuclear magnetic resonance (NMR) and scanning electron microscopy (SEM) techniques were used to obtain the microscopic features of the CPB specimens. The relationships between the permeability and uniaxial compressive strength (UCS) of the CPB specimens, and the microscopic characteristics of the CPB specimens, were established. The results showed the following: (i) The permeability of the CPB had an exponential functional relationship with its stone powder content. (ii) The stone powder content of CPB and the peak area of the $T_2$ spectrum are negatively correlated with the UCS. However, there was a $T_2$ peak area corresponding to the worst UCS with the same stone powder content. (iii) The peak area of the $T_2$ spectrum showed that the proportion of area of a small pore size was more than 80%, indicating that the pore size was mainly small. The pore diameter of small pores was linearly and inversely proportional to the UCS of the specimens. It can be found that the factors affecting the strength characteristics of CPB include not only the stone powder content, but also the curing conditions of different humidity.

# 1. Introduction

A filling mining method has been widely used in metal mining and is playing an important role in the development and use of mining resources. At present, the backfill used in the cemented filling mining method mainly uses tailings as the aggregate and cement as the cementing material [1]. The microstructure of such filling materials can affect the strength characteristics, and their interrelationships have led to extensive research in academia. Nuclear magnetic resonance (NMR) techniques can be used to study the distribution characteristics and evolution of water during the early hydration process of filling slurry and the variation law of the pore size distribution over time in cemented paste backfill (CPB) specimens [2,3]. Liu *et al.* [4] used NMR and scanning electron microscopy (SEM) to characterize the pore structure of backfill specimens and studied the relationship between the uniaxial compressive strength (UCS) and the pore characteristics of CPB. However, with the increase of various solid wastes and the high cost of cement, some scholars have done a lot of research on the disposal of wastes, which are mainly used to replace cement as a cementitious material [5,6]. However, this will reduce the strength of CPB. In order to increase the strength of CPB, a lot of research has been done [7–9].

The hydration of cement is affected by the level of humidity. In different stopes of underground mines, the environmental humidity is not the same. Humidity affects the hydration of CPB, which affects the strength characteristics of the CPB. There are some scholars who have studied the concrete curing humidity [10,11], but there are few studies on the curing humidity of CPB. Ma [12] found that, by studying the compressive strength of concrete specimens under constant temperature and different humidity conditions, it can be seen that the strength increases with an increase in the curing humidity. Under the different curing conditions of the mine, the influence in the hydration reaction of the backfill on the strength differs. Xiong [13] found that, with the slow increase of the downhole temperature, the compressive strength and elastic modulus of the backfill will gradually decrease. Xu *et al.* [14] found that, with an increase in temperature, the strength and elastic modulus of the cracked tailing backfill first increases and then decreases. However, they only studied the effect of temperature on the backfill and did not involve the effect of humidity. Gan *et al.* [15] found that, under the same curing age, the strength of the specimens in the downhole curing was lower than that under the standard curing conditions. However, the authors used too few control groups for the study on the curing humidity, and their research was not comprehensive.

In contrast with the above, the hydration reaction of cement in CPB under different humidity conditions, particularly the backfill in the replacement using stone powder cement, must have its own law. CPB specimens were prepared from tailings, stone powder and cement. After the same period (21 days) of curing, an impact test of the macroscopic and microscopic characteristics and the mechanical properties of the CPB specimens was carried out by curing the CPB specimens under different humidity environments for 7 days. NMR technology and a uniaxial compression test were used to explore the effects of humidity on the macroscopic and microscopic characteristics and mechanical laws of CPB.

# 2. Material and methods

## 2.1. Materials

The test materials included tailings as the aggregate and cement and stone powder as the cementitious materials. The experimental tailings were taken from mineral processing discharge products from the concentrator at Gaofeng Mine in Guangxi Province, China. The stone powder was taken from a quarry near the Gaofeng Mine in Guangxi Province, China. The basic properties of the above experimental materials were analysed using a laser particle sizer (Mastersizer 2000), X-ray fluorescence spectrometer and X-ray diffraction to determine the particle size distribution (figure 1) and physico-chemical properties (table 1). The cement was ordinary Portland cement, which was produced at the Changsha Xinxing Cement Factory. The chemical composition of cement is shown in table 2. Tap water was applied as the experimental water.

## 2.2. Experiment methods

### 2.2.1. Experimental design scheme

Tailings were used as backfill aggregates, and ordinary Portland cement and stone powder were used as the cementitious materials. In this study, stone powder cement tailings ratio (SPCTR) refers to the ratio of

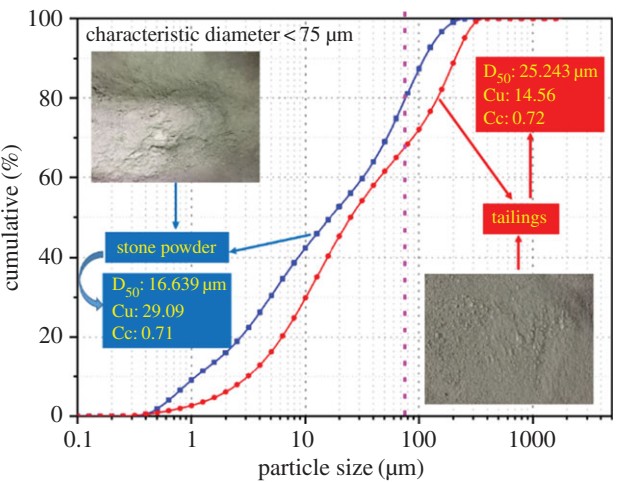

**Figure 1.** Particle size distribution.

**Table 1.** Physical properties and chemical composition of the tailings and stone powder.

| property | kind | contents | | | |
|---|---|---|---|---|---|
| physical properties | class | apparent density (g/cm$^3$) | | packing density (g/cm$^3$) | surface moisture content (%) |
| | tailings | 3.49 | | 1.24 | 0.128 |
| | stone powder | 2.89 | | 0.99 | 0.162 |

| chemical composition | chemical composition | O | Fe | S | Ca | Si | Mg | Al |
|---|---|---|---|---|---|---|---|---|
| | tailings (%) | 34.699 | 23.098 | 15.857 | 14.025 | 4.476 | 2.117 | 1.113 |
| | stone powder (%) | 58.301 | 0.942 | 0.298 | 23.279 | 14.901 | 0.471 | 1.273 |

**Table 2.** Chemical composition of cement.

| chemical composition | 3Ca · SiO$_2$ | 2Ca · SiO$_2$ | 3CaO · Al$_2$O$_3$ | 4CaO · Al$_2$O$_3$ · Fe$_2$O$_3$ |
|---|---|---|---|---|
| content (%) | 52.8 | 20.7 | 11.5 | 8.8 |

stone powder and cement to the tailings. The SPCTR and mass fractions (MFs) were set to 1 : 3 and 73%, respectively. A control experiment was conducted on different ratios of stone powder and cement. According to the research by Feng & Sun [16] and Liu [17], at 21 days, the hydration reaction of most of the cement in the cement-containing cementitious materials was completed owing to the influence of moisture in the material. After 21 days, a rehydration reaction of the material occurred owing to the influence of moisture in the environment. A scheme regarding the influence of environmental humidity on a cement rehydration reaction was designed. The design of this experiment includes an initial standard curing for 21 days followed by curing for 7 days under different humidity conditions.

The manufacturing and curing processes of the CPB specimens were as follows: Various raw materials were weighed according to the designed ratio and they were placed in a mixing bowl and mixed well by a mixer. Different ratio mortars were made into standard specimens using Φ50 mm × 100 mm cylindrical moulds. According to the content of the stone powder, the specimens were divided into eight groups (labelled as A1–A8) from low to high, and each ratio was 18 pieces, with 144 blocks in total. Each set of specimens was divided into six categories according to the curing and disposal conditions, with three pieces each, labelled C1–C6. They were initially co-cured for 21 days under 99% humidity and in a 20°C curing box. After 21 days of curing, they were processed according to the six design conditions. The specific curing conditions are shown in table 3. In table 3, A1–A8 of C5, and C6 were placed in the original curing box for 7 days of curing, and then those of

**Table 3.** Curing conditions of CPB specimens.

| kind group | 0% | 30% | 70% | 99% | saturated water treatment | saturated water treatment and centrifugation |
|---|---|---|---|---|---|---|
| A1 | C1A1(1)–C1A1(3) | C2A1(4)–C2A1(6) | C3A1(7)–C3A1(9) | C4A1(10)–C4A1(12) | C5A1(13)–C5A1(15) | C6A1(16)–C6A1(18) |
| A2 | C1A2(1)–C1A2(3) | C2A2(4)–C2A2(6) | C3A2(7)–C3A2(9) | C4A2(10)–C4A2(12) | C5A2(13)–C5A2(15) | C6A2(16)–C6A2(18) |
| A3 | C1A3(1)–C1A3(3) | C2A3(4)–C2A3(6) | C3A3(7)–C3A3(9) | C4A3(10)–C4A3(12) | C5A3(13)–C5A3(15) | C6A3(16)–C6A3(18) |
| A4 | C1A4(1)–C1A4(3) | C2A4(4)–C2A4(6) | C3A4(7)–C3A4(9) | C4A4(10)–C4A4(12) | C5A4(13)–C5A4(15) | C6A4(16)–C6A4(18) |
| A5 | C1A5(1)–C1A5(3) | C2A5(4)–C2A5(6) | C3A5(7)–C3A5(9) | C4A5(10)–C4A5(12) | C5A5(13)–C5A5(15) | C6A5(16)–C6A5(18) |
| A6 | C1A6(1)–C1A6(3) | C2A6(4)–C2A6(6) | C3A6(7)–C3A6(9) | C4A6(10)–C4A6(12) | C5A6(13)–C5A6(15) | C6A6(16)–C6A6(18) |
| A7 | C1A7(1)–C1A7(3) | C2A7(4)–C2A7(6) | C3A7(7)–C3A7(9) | C4A7(10)–C4A7(12) | C5A7(13)–C5A7(15) | C6A7(16)–C6A7(18) |
| A8 | C1A8(1)–C1A8(3) | C2A8(4)–C2A8(6) | C3A8(7)–C3A8(9) | C4A8(10)–C4A8(12) | C5A8(13)–C5A8(15) | C6A8(16)–C6A8(18) |

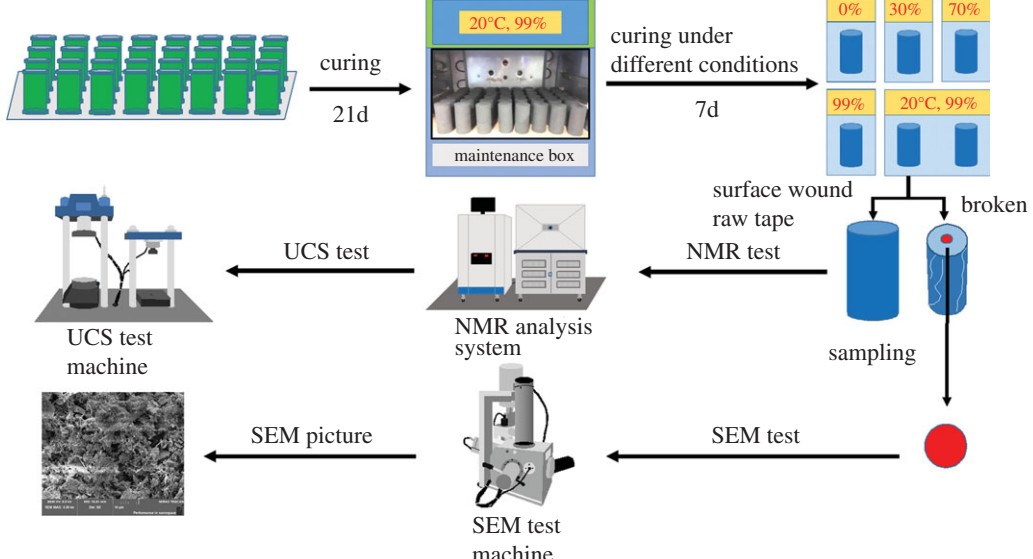

**Figure 2.** Experiment process with sample preparation, UCS tests and SEM.

**Table 4.** SPCTR, MFs, stone powder content (SP content), cement content and tailing content of CPB specimens.

| group | SPCTR | MF (%) | SP content/% | cement content (%) | tailing content (%) |
|---|---|---|---|---|---|
| A1 | 1:4 | 73 | 0.00 | 25.00 | 75 |
| A2 | | | 1.25 | 23.75 | 75 |
| A3 | | | 2.50 | 22.50 | 75 |
| A4 | | | 3.75 | 21.25 | 75 |
| A5 | | | 5.00 | 20.00 | 75 |
| A6 | | | 7.50 | 17.50 | 75 |
| A7 | | | 10.00 | 15.00 | 75 |
| A8 | | | 12.50 | 12.50 | 75 |

**Table 5.** NMR test parameter setting.

| parameter | SW (kHz) | SF (MHz) | O1 (Hz) | TD | DRG1 | NS | P1 (µs) | P2 (µs) | NECH | TW | RFD |
|---|---|---|---|---|---|---|---|---|---|---|---|
| number | 100 | 21 | 273173.80 | 2256 | 3 | 16 | 11.52 | 23.04 | 150 | 3000 | 0.3 |

C5 were subjected to saturated water treatment (recorded as C5A1–C5A8) and those of C6 were subjected to saturated water treatment and centrifugation (recorded as C6A1–C6A8). The curing temperature of all samples above is 20°C. The experimental ratios are shown in table 4.

### 2.2.2. Experimental test process

(1) NMR test: An NMR analysis was conducted using MesoMR23-060H. Parameter settings are shown in table 5. The cured specimens were tested by NMR. The NMR $T_2$ spectrum was obtained using NMR analysis software, and the area of the $T_2$ spectrum, the area of each peak and its proportion were calculated. Of these, the permeability parameter of samples subjected to saturation and centrifugation could be obtained by NMR.

(2) UCS test: The UCS test was carried out using a pressure tester No. 01000405 with a loading rate of 0.2 kN s$^{-1}$. The cured specimens were then placed in the middle of the pressure plate to be

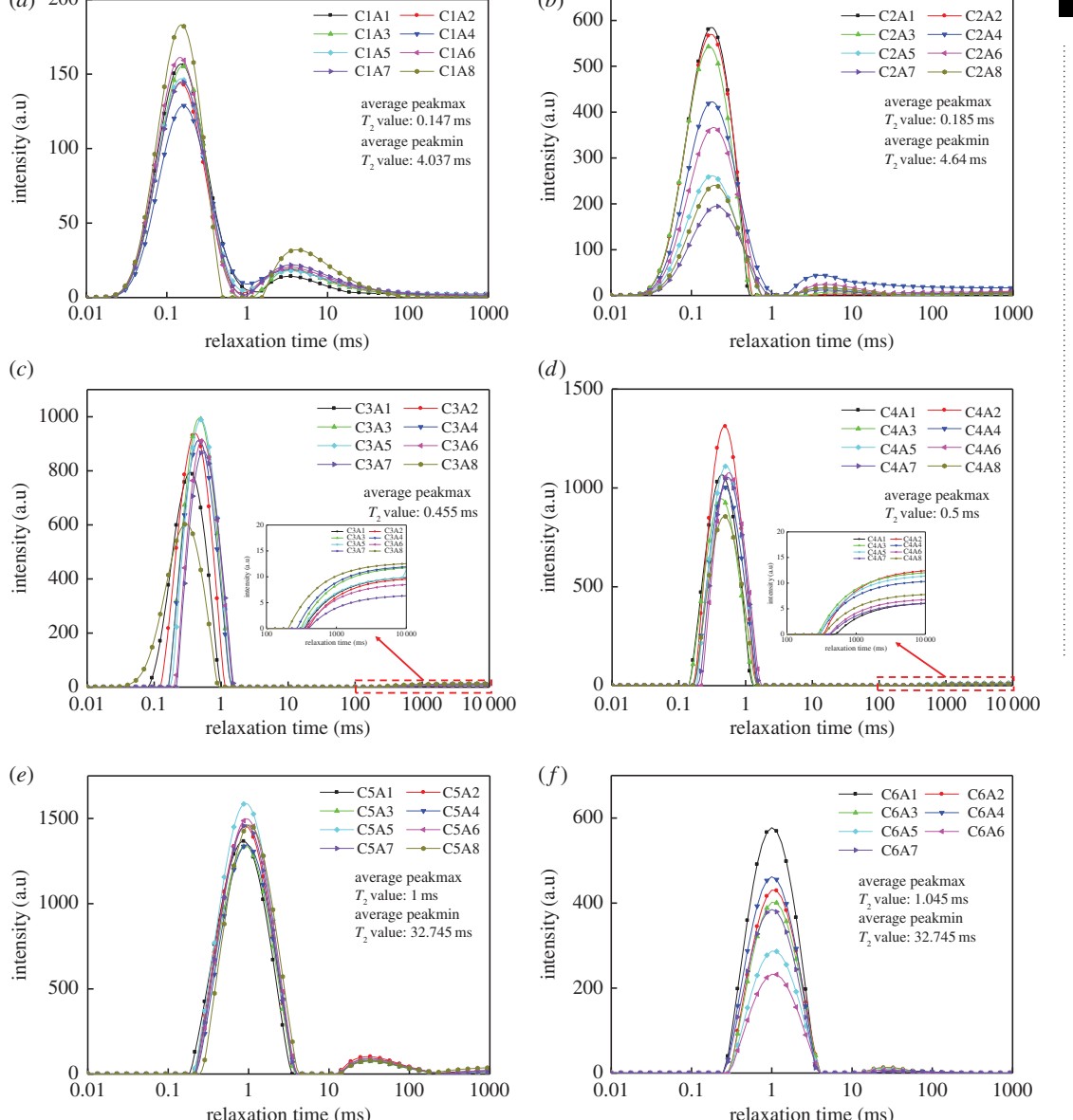

**Figure 3.** NMR $T_2$ spectrum distribution of CPB specimens under different curing conditions. (*a*) 0% humidity. (*b*) 30% humidity. (*c*) 70% humidity. (*d*) 99% humidity. (*e*) Saturated water. (*f*) Saturated water and centrifugation.

tested by the UCS test machine. The UCS value was obtained through a force measurement system analysis.

(3) SEM test: SEM was applied using a Quanta-200. We took cubic specimens with a side length of 5 mm from the centre of the specimens, and removed the dust from the surface; a metal conductive film was then applied on the vacuum coater, and finally a magnification of 5000x was set for electron microscopy to select the appropriate image. The experiment process is shown in figure 2.

# 3. Analysis results

## 3.1. NMR $T_2$ spectrum distribution

The distribution of the $T_2$ value represents the pore size distribution for samples which are saturated [18]. The transverse relaxation time $T_2$ of the NMR reflects the pore size; that is, the smaller the $T_2$ value, the smaller the pore diameter. Similarly, the larger the $T_2$ value, the larger the pore diameter. The number of pores that transverse the relaxation time of the peak value corresponds to the largest pore size [19]. The

**Table 6.** $T_2$ peak area and its proportion of CPB specimens under different curing conditions and stone powder content.

| curing conditions | group | first peak area | first peak area ratio (%) | second peak area | second peak area ratio (%) | total peak area |
|---|---|---|---|---|---|---|
| C1 | A1 | 117.0404 | 86.948 | 17.5693 | 10.139 | 134.6097 |
| | A2 | 114.1468 | 82.701 | 23.8767 | 17.299 | 138.0235 |
| | A3 | 112.1973 | 84.811 | 20.0936 | 15.189 | 132.2909 |
| | A4 | 104.9372 | 79.551 | 26.9747 | 20.449 | 131.9119 |
| | A5 | 103.9769 | 80.968 | 24.4404 | 19.032 | 128.4173 |
| | A6 | 102.9564 | 81.588 | 23.2342 | 18.412 | 126.1906 |
| | A7 | 101.7290 | 80.123 | 25.2370 | 19.877 | 126.9660 |
| | A8 | 99.9935 | 81.082 | 23.3305 | 18.918 | 123.3240 |
| C2 | A1 | 388.2096 | 96.973 | 12.1179 | 3.027 | 400.3275 |
| | A2 | 387.6869 | 95.127 | 19.8598 | 4.873 | 407.5467 |
| | A3 | 365.4683 | 95.090 | 18.8711 | 4.910 | 384.3394 |
| | A4 | 327.1894 | 81.573 | 73.9108 | 18.427 | 401.1002 |
| | A5 | 282.3138 | 91.518 | 26.1652 | 8.482 | 308.4790 |
| | A6 | 268.6997 | 88.467 | 35.0291 | 11.533 | 303.7288 |
| | A7 | 145.5212 | 87.385 | 21.0076 | 12.615 | 166.5288 |
| | A8 | 135.9360 | 89.458 | 16.0191 | 10.542 | 151.9551 |
| C3 | A1 | 577.1534 | 99.037 | 5.6120 | 0.963 | 582.7654 |
| | A2 | 527.4483 | 98.133 | 10.0348 | 1.867 | 537.4831 |
| | A3 | 550.7071 | 97.403 | 14.6832 | 2.597 | 565.3903 |
| | A4 | 512.7181 | 97.907 | 10.9606 | 2.093 | 523.6787 |
| | A5 | 524.8990 | 97.910 | 11.2045 | 2.090 | 536.1035 |
| | A6 | 482.6831 | 98.786 | 5.9318 | 1.214 | 488.6149 |
| | A7 | 460.8953 | 98.923 | 5.0178 | 1.077 | 465.9131 |
| | A8 | 413.6614 | 98.215 | 7.5181 | 1.785 | 421.1795 |
| C4 | A1 | 783.3576 | 97.738 | 18.1297 | 2.262 | 801.4873 |
| | A2 | 728.0665 | 98.113 | 14.0028 | 1.887 | 742.0693 |
| | A3 | 634.1286 | 97.603 | 15.5733 | 2.397 | 649.7019 |
| | A4 | 561.0880 | 97.259 | 15.8129 | 2.741 | 576.9009 |
| | A5 | 622.6891 | 97.990 | 12.7728 | 2.010 | 635.4619 |
| | A6 | 570.6753 | 98.108 | 11.0054 | 1.892 | 581.6807 |
| | A7 | 549.6662 | 98.563 | 8.0139 | 1.437 | 557.6801 |
| | A8 | 455.7898 | 96.081 | 18.5910 | 3.919 | 474.3808 |

NMR $T_2$ spectrum distribution of the CPB specimens is shown in figure 3 under different humidity curing conditions and stone powder content. As shown in figure 3, the two peaks were found in the $T_2$ spectrum distribution of the test specimens. From figure 3a–f, the relaxation times of the first peak value are 0.15, 0.18, 0.46, 0.49, 1 and 1.14 ms, respectively. In figure 3a,b,e and f, the relaxation times of the second peak are 3.51, 4.04, 32.74 and 32.74 ms, respectively. The second peak values of figure 3c,d are smaller, and the transverse relaxation time, $T_2$, is large. This indicates that, after 21 days, a rehydration reaction had taken place. The higher the curing humidity, the larger the most probable pore size of the small pores. It was proved that the greater the environmental humidity, the more capillary water occurring in the CPB specimens, providing more moisture to accelerate the cement rehydration reaction. The cement rehydration reaction was accelerated, resulting in numerous porous and large-pore-sized hydration products. Under the same humidity condition, the first peak of

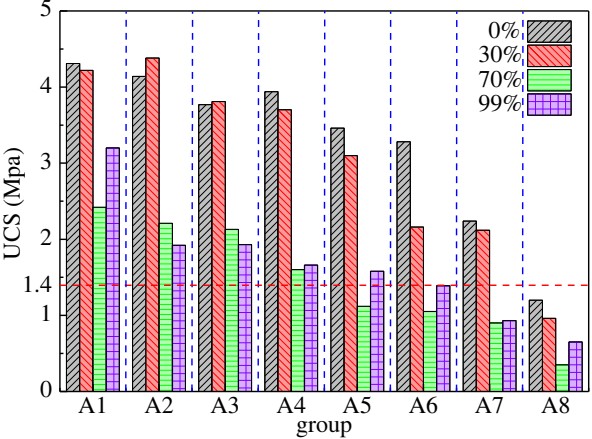

**Figure 4.** UCS of CPB specimens under different curing conditions and stone powder content.

all specimens was significantly higher than the second peak, indicating the largest number of small pores in the specimens. Under dry conditions, the first peak is between 100 and 200 a.u.; under 30% humidity, the first peak value is between 200 and 600 a.u.; under 70% humidity, the first peak value is between 600 and 1000 a.u.; under 99% humidity, the first peak value is between 800 and 1300 a.u.; whereas after water saturation, the first peak value is between 1300 and 1600 a.u.; and after water saturation and centrifugation, the first peak value is between 200 and 600 a.u. This indicates that the greater the humidity, the greater the number of small pores. There are two reasons for this. The first is that the greater the environmental humidity, the more capillary water and gel pore water that occurs in the CPB specimens, resulting in an increase in the water content of the small pores in the specimens. The second is that the greater the humidity, the faster the cement hydration reaction. The hydration product fills the pores of the specimens, resulting in smaller pores. The first peak value after centrifugation is not significantly different from the first peak value under 30% humidity, whereas the second peak value is smaller than the second peak value under 30% humidity, indicating that centrifugation not only caused all pores to lose moisture but also caused large pores to lose more moisture.

## 3.2. $T_2$ spectrum peak area

The area of the NMR $T_2$ spectrum is in direct ratio with the water content in the specimens [4,19]. Table 6 shows the total peak area of the $T_2$ spectrum, the area of each peak and its proportion in each specimen under different humidity curing conditions and stone powder content. It can be seen that the higher the content of stone powder in the specimens, the smaller the total area of the $T_2$ spectrum under the same humidity. This indicates that an increase in the stone powder amount in the specimens leads to an accelerated cement hydration reaction, which consumes more water in the specimens, resulting in a decrease in their moisture content. Under the same stone powder amount, the greater the curing humidity, the larger the total area of the $T_2$ spectrum of the specimens. This indicates that the more environmental moisture that occurs, the more pore water in the specimens. The amount of water consumed by the cement hydration reaction is lower than the increase in pore water, resulting in an increase in water content in the specimens. The first peak area was between 80 and 99% of the total area, indicating that the pores in the specimens were mostly small. This occurs because the cement hydration reaction produces many fine hydration products, filling in the pores of the specimens, resulting in smaller pores (which occurs because, in cement hydration, the pores are filled by fine hydration products). Under the same humidity, the proportion of the first peak area decreased with the increase in stone powder content. The proportion of the second peak area increased with an increase in the stone powder content. This indicates that the consumption of the capillary water is larger than that of the macroporous water, which proves that the increase in the stone powder content in the specimens will accelerate the cement hydration reaction. Cai [20] also found a phenomenon in that stone powder will accelerate the cement hydration reaction when studying stone powder concrete. Under the same powder content, from 0 to 70% humidity, the proportion of the first peak area gradually increased, and the proportion of the second peak area gradually decreased. This indicates that, with an increase in humidity, the water evaporated in the large pores consumes more

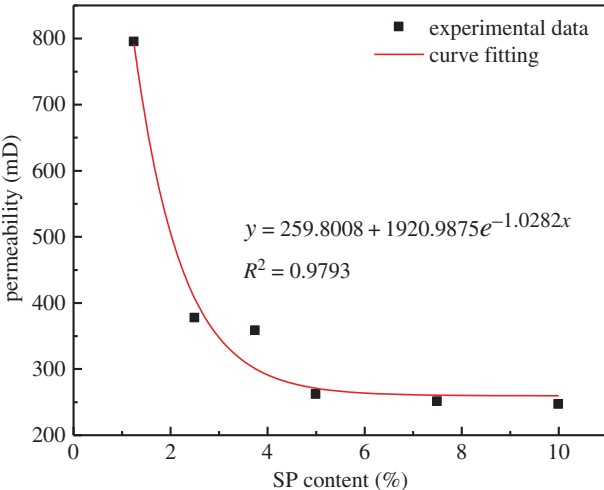

**Figure 5.** Relationship between permeability and stone powder amount of CPB specimens.

water than the cement hydration reaction in the capillaries. However, when the humidity reaches 99%, the proportion of the first peak area is not increased, but is rather decreased, and the proportion of the second peak area is not reduced but increased. This shows that, when the environment reaches a certain level of humidity, the evaporation of water in the large holes will slow down. A4 under 30% humidity, A3 under 70% humidity and A4 under 99% humidity showed some abnormalities.

## 3.3. Strength characteristics

Figure 4 shows the UCS of specimens with eight different stone powder contents under four different humidity curing conditions. A pressure of 1.4 MPa is the basic requirement to meet the strength of the mine (Gaofeng Mine) backfill [21]. Because the strength of the saturated water specimens was too low, the strength was not tested. It can be seen from figure 4 that, under the same humidity conditions, the strength tends to decrease as the amount of stone powder increases. Li [22] also found a similar phenomenon in that excessive stone powder content leads to a reduction in strength when studying stone powder concrete. There are two reasons for this phenomenon. On the one hand, the stone powder absorbs some of the free water and reduces the amount of cement slurry in the specimens, resulting in a decrease in the adhesion between aggregates. On the other hand, the increase in the amount of stone powder in the specimens causes the gradation to become unreasonable, resulting in a weakening of the skeleton effect of the cement slurry and the aggregate. Clearly, the greater the humidity, the lower the strength because, when the humidity is too high, more moisture occurs between the particles. A disassembly pressure was generated, and the force of bonding matter was reduced; thus, the CPB was loosened, reducing the UCS of the CPB. When the stone powder content is the same, the UCS decreases as obviously as the humidity increases. It shows that the moist environment seriously affects the UCS. Although the moist environment could promote the hydration reaction, at this time, the moisture content is a decisive factor affecting the strength, which results in a decrease in the strength of CPB.

# 4. Discussion and analysis

## 4.1. Permeability

Permeability is a physical quantity used to characterize the ability of porous media materials to transport liquids, obtained by Coates model in NMR, the size of which determines the quality of the permeability. The permeability of the backfill is directly related to the water secretion ability of the filling slurry, and is one of the important properties of the backfill. To be able to drain the moisture in the backfill faster, reduce the stopping cycle and reduce the pressure on the retaining wall and filtering well, the mine needs a suitable permeability of the backfill [23]. Therefore, it is important to study the permeability. Here, the permeability of the hardened specimens was obtained using saturated water and

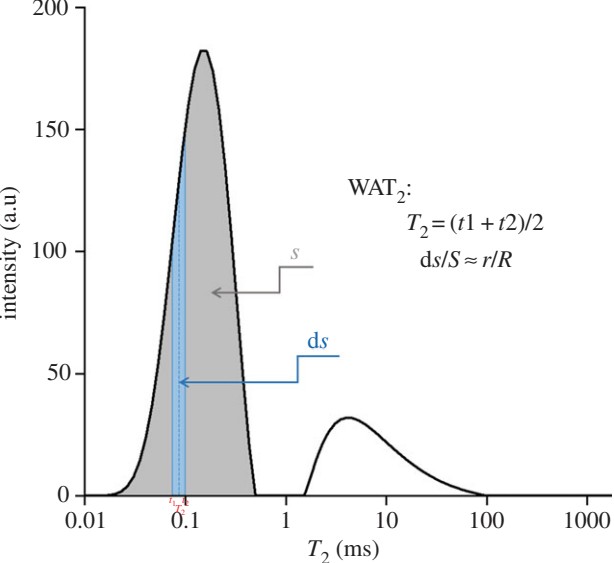

**Figure 6.** WAT$_2$ schematic diagram.

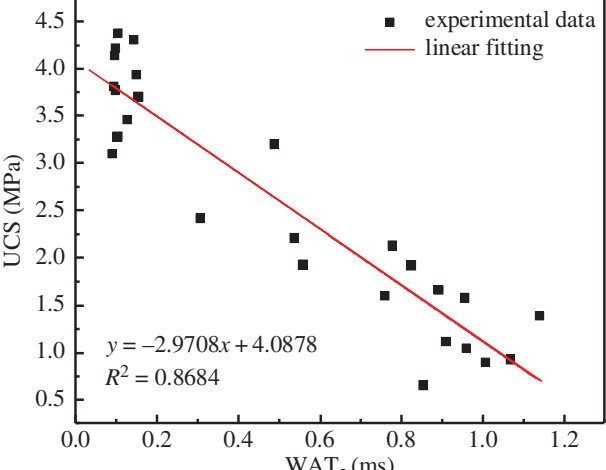

**Figure 7.** Relationship between UCS and WAT$_2$ value of cemented backfill specimens.

centrifugation processing (i.e. C6A1–C6A8). Figure 5 shows the relationship between the permeability of the CPB specimens and the amount of stone powder. The permeability of the CPB specimens exponentially decreases with the amount of stone powder. This relationship is as follows:

$$y = 259.8008 + 1920.9875e^{-1.0282x}, \tag{4.1}$$

where $y$ represents the permeability (mD), and $x$ represents the amount of stone powder in the cemented backfill specimens (%).

There are two reasons for this relationship. First, the stone powder can accelerate the hydration reaction of the cement to form finer products, and thus the porous medium contained finer particles, the internal structure became denser and the porosity became smaller, resulting in a poor permeability of the backfill. Second, the stone powder particles were small, filling in the voids in the specimens and reducing the porosity, resulting in a poor permeability of the CPB.

## 4.2. Relationship between UCS and pore size distribution

Regarding the value of UCS, many factors affect the UCS of the CPB. The pore size in the CPB was one of the influencing factors [24]. In §3.2, it was mentioned that the first peak area accounted for more than 80% of the total area. Here, the influence of the pore size of the small pores on the strength of the

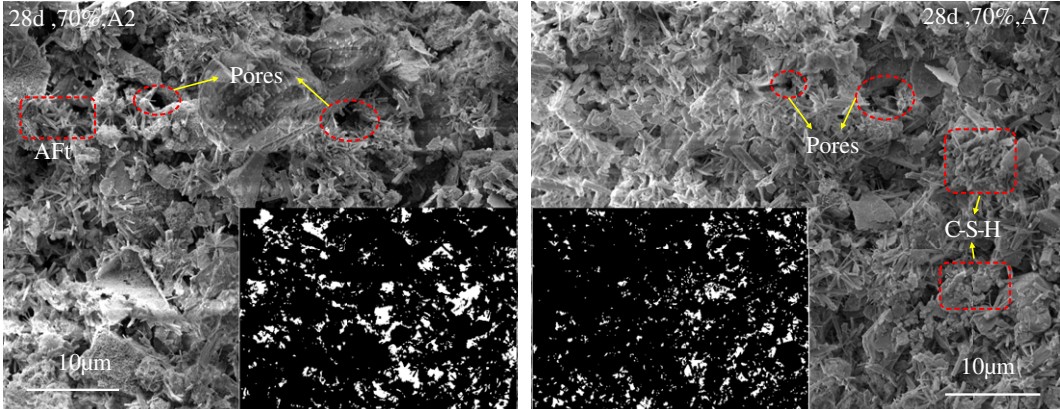

**Figure 8.** Magnified SEM image.

specimens was studied. For the pores of the spherical model and tube bundle model, the relationship between the pore radius r and the transverse relaxation time $T_2$ can be approximated as

$$\frac{1}{T_2} \approx \rho_2\left(\frac{S}{V}\right) = \frac{F_s\rho_2}{r},\tag{4.2}$$

where $T_2$ represents the total transverse relaxation time (ms), $r$ represents the pore radius (μm), $F_s$ represents the pore shape factor (for spherical pores, $F+ = 3$, and for tube bundle pores, $F_s = 2$), and $\rho_2$ represents the transverse surface relaxation strength ($\mu m\,ms^{-1}$), S represents the pore surface area ($cm^2$), V represents the pore volume ($cm^3$).

So according to equation (4.2), the pore radius could be expressed as

$$r \approx F_s\rho_2 T_2 .\tag{4.3}$$

The peak area of the first peak in the $T_2$ spectrum has a direct ratio with the water amount of the pores. The water amount of the pores is a direct ratio with the number of small pores, and thus the peak area of the first peak reflects the number of small pores. The ratio of the $T_2$ spectral area ($ds$) between the two adjacent $T_2$ values ($t_1$ and $t_2$) in the first peak of the $T_2$ spectrum and the total area ($S$) of the first peak reflects the ratio of the pore size ($r$) corresponding to the intermediate $T_2$ ($T_2$) value of the above two adjacent $T_2$ values to the total number of small pores ($R$). $WAT_2$ schematic diagram is shown in figure 6. Using the above ratio, the weighted average transverse relaxation time ($WAT_2$) is calculated based on the weighted average of the transverse relaxation time of the first peak of the $T_2$ spectrum, and the relationship between the pore size and the UCS is then obtained using $WAT_2$. The relationship between $WAT_2$ and the UCS of the specimens is shown in figure 7. It can be seen that $WAT_2$ is linearly and inversely proportional to the UCS of the specimens, and its relationship is as follows:

$$y = -2.9708x + 4.0878,\tag{4.4}$$

where $y$ represents UCS (MPa) and $x$ represents $WAT_2$ (ms).

As $WAT_2$ increases, the UCS decreases. This phenomenon also occurred when Zhu *et al*. [25] studied silicon powder cement. The larger the $WAT_2$, the larger the pore size and the worse the compactness of the backfill, resulting in a decrease in the UCS of the specimens. Figure 7a,b show SEM images (10 000×) for the A2 and A7 specimens under 70% humidity, respectively. The binarization image can well reflect the distribution of pores. Binarization mainly uses threshold greyscale segmentation to transform the solid phase and pores in the image into black and white regions. The principle is

$$f(i,j) = \begin{cases} 0, f(i,j) < T, \\ 1, f(i,j) \geq T, \end{cases}$$

where $T$ is the threshold grey scale. By the binarization function, the pixel points, whose grey values are lower than $T$, are set to 0, that is, the black point (solid phase) and the pixel points, whose grey values are higher than or equal to $T$, are set to 1, that is, white point (pore).

It can be seen from figure 8 that the CPB is formed by a large number of acicular substances forming a loose network structure, such that a large number of pores appear, and the pores are distributed in different shapes and quantities and the pore size distribution is not uniform [26], the ability for the

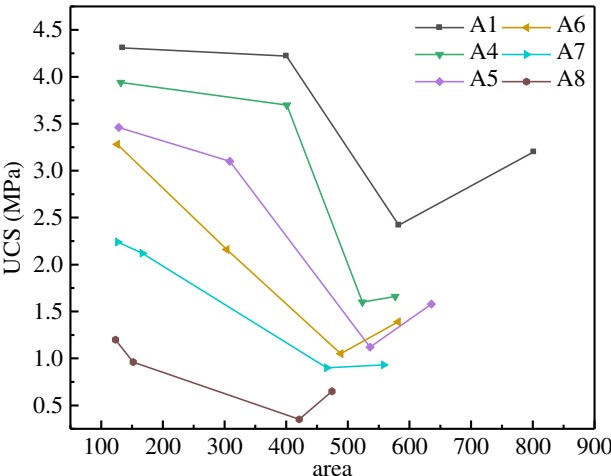

**Figure 9.** Relationship between UCS and $T_2$ peak area under different humidities.

pore structure to resist external forces was reduced. The larger the pore size, the lower the resistance; the looser the structure, the worse the compactness of the CPB and the lower the strength in terms of the macro-performance. By comparing the two images, it can be seen that the higher the stone powder content, the smaller the pore size. This occurs because the stone powder particles are small and fill in the pores in the specimens, resulting in smaller pores. This indicates that the stone powder can refine the pore structure to a certain extent.

## 4.3. Relationship between UCS and NMR $T_2$ spectrum area

The area of the NMR $T_2$ spectrum can reflect the amount of fluid contained in the specimens [18]. Figure 9 shows the relationship between the $T_2$ spectral area and the UCS. It can be seen from the figure that under the same amount of stone powder, the UCS decreases initially and then increases with the increase in the spectral area. This indicates that excessive humidity is not necessarily harmful to the strength. The declining trend in the front section occurs because the hydration products of the previous cement fill in most of the pores, and no more space is available to accommodate the new hydration products produced during the later stage. However, as the amount of water in the specimens increases, the cement hydration reaction still accelerates. The hydration product causes pressure on the CPB to reduce the strength. Feng *et al.* [27] also found a similar phenomena when studying concrete. An increasing trend at the back end indicated that carbonic acid in the moisture reacted with the hydration product CH: $Ca(OH)_2 + CO_2 + H_2O \rightarrow CaCO_3 + 2H_2O$ [28]. This occurred because the water in the specimens came from tap water, which contained $CO_2$ and reacted with water to form carbonic acid. As the moisture content in the specimens increased, the amount of carbonic acid also increased. When a certain amount of carbonic acid was reached, the carbonic acid reacted with the hydrated product CH, reducing the CH in the specimens to increase the strength. In addition, when the content of the stone powder was the same, there was a $T_2$ peak area with the worst UCS, which indicated that, when the water content reached a certain amount, the cement hydration reaction rate reached the maximum and the hydrated product had the fastest production rate and the largest pore diameter, resulting in the lowest strength.

## 5. Conclusion

There are many factors that affect the strength of CPB, and the curing environment with different humidity is one of them. In order to analyse the strength characteristics and microstructure of CPB, the pore characteristics, microstructure and strength characteristics of CPB were tested using NMR, SEM and a single-axis compressor in this study. The permeability, UCS, relationship between the UCS and pore size and the moisture content of the specimens were analysed.

(1) There are two peaks shown in the NMR $T_2$ spectrum distribution. The first peak area accounts for 80–99% of the total area. The first peak is significantly higher than the second peak, indicating

that the specimens are mostly small pores. As the curing humidity increases, the $T_2$ value of the peak and the proportion of the area of the first peak all gradually increase, indicating that the pore size and proportion of small pores increase. Under the same curing humidity conditions, the UCS of the specimens decreases with an increase in the amount of stone powder. It was found through an analysis that the UCS of the specimens is linearly and inversely proportional to the pore size of the small pores, the main reason for which is that both the hydration product and the stone powder change the pore size, thus changing the compactness of the CPB.

(2) As the amount of stone powder in the specimens increases, the internal structure of the specimens becomes denser and the porosity becomes smaller, resulting in a decrease in exponential permeability.

(3) Under the same stone powder content, the strength of the specimens first decreases and then increases with an increase in the water content. The main reason for the turning of the specimen strength is the consumption of the CH in the hydration product and conversion into $CaCO_3$.

Although this research has achieved some outcomes, there are still some limitations, such as various measurement errors. And this study needs further research, such as the co-effect of temperature and humidity changes on CPB. However, this research is still very meaningful. The use of stone powder instead of cement can effectively reduce the filling cost. The research on the influence of curing humidity on CPB can provide guidance for CPB in different stopes to meet the requirements of mines.

Ethics. This work having obtained permission from all the authors, we declare that the present experiments in the manuscript were performed in accordance with the standard of academic conduct from the Chinese academic society. All relevant ethical safeguards have been met in relation to patient or subject protection or animal experimentation.

Data accessibility. The datasets supporting this article have been uploaded as part of the electronic supplementary material.

Authors' contributions. J.H. and Q.R. designed the experimental process. Z.L. and F.Z. prepared the stone powder cement tailings backfill samples. Y.K. and T.Z. tested strength of all samples and finished other tests. Q.R. and F.Z. collected and analysed the data. J.H. and Q.R. interpreted the results and wrote the manuscript.

Competing interests. The authors declare no conflict of interest.

Funding. The research was supported by (i) Ministry of Science and Technology of the People's Republic of China (grant no. 2017YFC0602901); (ii) National Natural Science Foundation of China (grant no. 41672298).

Acknowledgements. We thank the Gaofeng Mine's management and staff for their valuable support. We thank instructional support specialist Modern Analysis and Testing Central of Central South University.

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
