## [Reviewer comments · Royal Society Open Science]

Review History

RSOS-191227.R0 (Original submission)

Review form: Reviewer 1

Is the manuscript scientifically sound in its present form?

Yes

Are the interpretations and conclusions justified by the results?

Yes

Is the language acceptable?

Yes

Do you have any ethical concerns with this paper?

No

Have you any concerns about statistical analyses in this paper?

No

Recommendation?

Accept with minor revision (please list in comments)

Comments to the Author(s)

Reviewer's comments:

Review report for Royal Society Open Science (RSOS-191227): Microscopic characterization and strength characteristics of cemented backfill under different humidity curing conditions

This paper presents the results of a comprehensive experimental work considered for investigating the effect of different humidity conditions on mechanical and microstructural properties of cemented paste tailings as a function of different curing times. The topic addressed in this work is interesting and within the scope of the journal. The experimental program is well designed and executed. The authors have obtained good results. The paper's structure and technical quality are fairly good. I strongly believe that the presented data are useful to other researchers. This reviewer has some minor corrections and/or comments which can improve the paper's quality. All comments/suggestions are as follows;

Specific comments

Abstract

...standard curing...

Please explain this in more detail in this section.

What is the main outcome of this research? Please give it at the end of the abstract section.

Introduction

There are lots of works already published in the literature which deal with the effect of internal and external factors on performance and quality of cemented paste backfill. Please consider the following works in this study: Xue et al., 2019a,b,c; Cao et al., 2018a,b, 2019; Yilmaz 2011; 2018; Yilmaz et al., 2013.

Please rewrite the introduction section by focusing on problematic, suggested methods, and originality and specific objectives. Please use the present literature critically, not giving their whole details.

At the end of this section, please provide the originality and specific objectives of this study.

Materials and Methods

Please provide further information on tailings samples. Where the authors collect these samples in the mine? They come from tailings dams or paste plant filtering cakes or mineral processing discharge products. Please state it clearly.

What were the sample preparation and mixing conditions? Please mention about them shortly.

How is the manufacturing and curing process of CPB samples? Please make it simple in the revised paper.

Please explain the experimental test procedures in more detail. How were the working conditions of samples? Please state them clearly in the revised paper.

Please add the permeability tests in this section since the authors provide some results on this test later on.

Analysis results

Are there any threshold limits of NMR results? If yes, please discuss it in the body of the manuscript as well as in Figure 3 (as a dotted line).

What are the main reasons for observing this behavior in Figure 4 in terms of strength differences? Please discuss it thoroughly.

Results and discussion

Which type of permeability tests were followed in this study? Please state it clearly.

In Figure 5; permeability tests were conducted on fresh or hardened samples? Please clear it in the revised paper.

In Figure 7; this figure is very nice. The authors need to present in a larger size for better resolution.

Conclusion

Please provide a short summary before highlighting the main findings of this research project.

What are the limitations and recommendations of this work which can give some inspirations to other researchers to be worked in the area of this research?

As a reviewer, I am pleased to suggest the paper to be accepted for publication in *Construction and Building Materials* through the aforementioned minor revisions.

References

- Cao, S., Song, W., Yilmaz, E., 2018a. Influence of structural factors on uniaxial compressive strength of cemented tailings backfill. *Construction and Building Materials*, 174, 190-201.
- Cao, S., Yilmaz, E., Song, W., 2018b. Dynamic response of cement-tailings matrix composites under SHPB compression load. *Construction and Building Materials*, 186, 892-903.
- Cao, S., Yilmaz, E., Song, W., 2019. Fiber type effect on strength, toughness, and microstructure of early age cemented tailings backfill. *Construction and Building Materials*, 223, 44-54.
- Xue, G., Yilmaz, E., Song, W., Cao, S., 2019b. Analysis of internal structure behavior of fiber-reinforced cement-tailings matrix composites through X-ray computed tomography. *Composites Part B: Engineering*, 175, 107091.
- Xue, G., Yilmaz, E., Song, W., Cao, S., 2019c. Mechanical, flexural and microstructural properties of cement-tailings matrix composites: Effect of fiber type and dosage. *Composites Part B: Engineering*, 172, 131-142.
- Xue, G., Yilmaz, E., Song, W., Yilmaz, E., 2019a. Influence of fiber reinforcement on mechanical behavior and microstructural properties of cemented tailings backfill. *Construction and Building Materials*, 213, 275-285.
- Yilmaz, E., 2011. Advances in reducing large volumes of environmentally harmful mine waste rocks and tailings. *Mineral Resources Management*, 27 (2), 89-112.
- Yilmaz, E., Belem, T., Benzaazoua, M., 2013. Study of physicochemical and mechanical characteristics of consolidated and unconsolidated cemented paste backfills. *Mineral Resources Management*, 29(1), 81-100.
- Yilmaz, E., 2018. Stope depth effect on-field behavior and performance of cemented paste backfills. *International Journal of Mining, Reclamation, and Environment*, 32(4), 273-296.

Review form: Reviewer 2

Is the manuscript scientifically sound in its present form?

Yes

Are the interpretations and conclusions justified by the results?

Yes

Is the language acceptable?

Yes

Do you have any ethical concerns with this paper?

No

Have you any concerns about statistical analyses in this paper?

No

Recommendation?

Major revision is needed (please make suggestions in comments)

Comments to the Author(s)

The manuscript explored the characteristics of cemented backfill under different humidity curing conditions using NMR and SEM techniques, which contributes to the existing literatures. It is an interesting research that deserves to be published. However, some comments should be addressed as follows:

- 1 The introduction part needs to be extended. The Introduction should present the reasons why studying the effect of humidity in CPB is necessary, rather than just list previous publications in the related field.
- 2 The references should be updated. The authors are suggested to cite some latest literatures, as the CPB develops rapidly.
- 3 WAT2 is unclear, and it should be explained with a schematic diagram.
- 4 The sum of element contents isn't 100% In table2?
- 5 The NMR parameters are important in this manuscript and are not mentioned in the paper. The authors are suggested to provide the NMR parameters.
- 6 The authors claimed that "The distribution of the T2 value represents the pore size distribution", which is not suitable for some samples in this paper are not saturated.
- 7 Please specify the application conditions of Equation 2.

Review form: Reviewer 3

Is the manuscript scientifically sound in its present form?

Yes

Are the interpretations and conclusions justified by the results?

Yes

Is the language acceptable?

Yes

Do you have any ethical concerns with this paper?

No

Have you any concerns about statistical analyses in this paper?

No

Recommendation?

Accept with minor revision (please list in comments)

Comments to the Author(s)

(1) What is the chemical composition of cement ? and it would be better to explain the derivation process of Equation.

(2) what is the curing temperature of all samples?

(3) which mine's UCS basic requirement is 1.4MPa, and what are the meanings of white areas and black areas in SEM binary images ?

Decision letter (RSOS-191227.R0)

18-Sep-2019

Dear Dr Zhao,

The editors assigned to your paper ("Microscopic characterization and strength characteristics of cemented backfill under different humidity curing conditions") have now received comments from reviewers. We would like you to revise your paper in accordance with the referee and Associate Editor suggestions which can be found below (not including confidential reports to the Editor). Please note this decision does not guarantee eventual acceptance.

Please submit a copy of your revised paper before 11-Oct-2019. Please note that the revision deadline will expire at 00.00am on this date. If we do not hear from you within this time then it will be assumed that the paper has been withdrawn. In exceptional circumstances, extensions may be possible if agreed with the Editorial Office in advance. We do not allow multiple rounds of revision so we urge you to make every effort to fully address all of the comments at this stage. If deemed necessary by the Editors, your manuscript will be sent back to one or more of the original reviewers for assessment. If the original reviewers are not available, we may invite new reviewers.

- Data accessibility

If you wish to submit your supporting data or code to Dryad (<http://datadryad.org/>), or modify your current submission to dryad, please use the following link:
<http://datadryad.org/submit?journalID=RSOS&manu=RSOS-191227>

- Competing interests

- Authors' contributions

- Acknowledgements

- Funding statement

Kind regards,

Lianne Parkhouse
Royal Society Open Science
openscience@royalsociety.org

on behalf of the Associate Editor and Professor R. Kerry Rowe (Subject Editor)
openscience@royalsociety.org

Associate Editor's comments:

Three reviewers have commented, with each providing at least some suggestions for improvement in the manuscript. Given the volume of the requested changes, we would like you revise the manuscript, and will invite the reviewers to look once again at the paper before a final decision is rendered.

Reviewers' Comments to Author:

Reviewer: 1

Review report for Royal Society Open Science (RSOS-191227): Microscopic characterization and strength characteristics of cemented backfill under different humidity curing conditions

This paper presents the results of a comprehensive experimental work considered for investigating the effect of different humidity conditions on mechanical and microstructural properties of cemented paste tailings as a function of different curing times. The topic addressed in this work is interesting and within the scope of the journal. The experimental program is well designed and executed. The authors have obtained good results. The paper's structure and technical quality are fairly good. I strongly believe that the presented data are useful to other researchers. This reviewer has some minor corrections and/or comments which can improve the paper's quality. All comments/suggestions are as follows;

Specific comments

Abstract

...standard curing...

Please explain this in more detail in this section.

What is the main outcome of this research? Please give it at the end of the abstract section.

Introduction

There are lots of works already published in the literature which deal with the effect of internal and external factors on performance and quality of cemented paste backfill. Please consider the following works in this study: Xue et al., 2019a,b,c; Cao et al., 2018a,b, 2019; Yilmaz 2011; 2018; Yilmaz et al., 2013.

Please rewrite the introduction section by focusing on problematic, suggested methods, and

originality and specific objectives. Please use the present literature critically, not giving their whole details.

At the end of this section, please provide the originality and specific objectives of this study.

Materials and Methods

Please provide further information on tailings samples. Where the authors collect these samples in the mine? They come from tailings dams or paste plant filtering cakes or mineral processing discharge products. Please state it clearly.

What were the sample preparation and mixing conditions? Please mention about them shortly.

How is the manufacturing and curing process of CPB samples? Please make it simple in the revised paper.

Please explain the experimental test procedures in more detail. How were the working conditions of samples? Please state them clearly in the revised paper.

Please add the permeability tests in this section since the authors provide some results on this test later on.

Analysis results

Are there any threshold limits of NMR results? If yes, please discuss it in the body of the manuscript as well as in Figure 3 (as a dotted line).

What are the main reasons for observing this behavior in Figure 4 in terms of strength differences? Please discuss it thoroughly.

Results and discussion

Which type of permeability tests were followed in this study? Please state it clearly.

In Figure 5; permeability tests were conducted on fresh or hardened samples? Please clear it in the revised paper.

In Figure 7; this figure is very nice. The authors need to present in a larger size for better resolution.

Conclusion

Please provide a short summary before highlighting the main findings of this research project.

What are the limitations and recommendations of this work which can give some inspirations to other researchers to be worked in the area of this research?

As a reviewer, I am pleased to suggest the paper to be accepted for publication in *Construction and Building Materials* through the aforementioned minor revisions.

References

- Cao, S., Song, W., Yilmaz, E., 2018a. Influence of structural factors on uniaxial compressive strength of cemented tailings backfill. *Construction and Building Materials*, 174, 190-201.
- Cao, S., Yilmaz, E., Song, W., 2018b. Dynamic response of cement-tailings matrix composites under SHPB compression load. *Construction and Building Materials*, 186, 892-903.
- Cao, S., Yilmaz, E., Song, W., 2019. Fiber type effect on strength, toughness, and microstructure of early age cemented tailings backfill. *Construction and Building Materials*, 223, 44-54.

- Xue, G., Yilmaz, E., Song, W., Cao, S., 2019b. Analysis of internal structure behavior of fiber-reinforced cement-tailings matrix composites through X-ray computed tomography. *Composites Part B: Engineering*, 175, 107091.
- Xue, G., Yilmaz, E., Song, W., Cao, S., 2019c. Mechanical, flexural and microstructural properties of cement-tailings matrix composites: Effect of fiber type and dosage. *Composites Part B: Engineering*, 172, 131-142.
- Xue, G., Yilmaz, E., Song, W., Yilmaz, E., 2019a. Influence of fiber reinforcement on mechanical behavior and microstructural properties of cemented tailings backfill. *Construction and Building Materials*, 213, 275-285.
- Yilmaz, E., 2011. Advances in reducing large volumes of environmentally harmful mine waste rocks and tailings. *Mineral Resources Management*, 27 (2), 89-112.
- Yilmaz, E., Belem, T., Benzaazoua, M., 2013. Study of physicochemical and mechanical characteristics of consolidated and unconsolidated cemented paste backfills. *Mineral Resources Management*, 29(1), 81-100.
- Yilmaz, E., 2018. Stope depth effect on-field behavior and performance of cemented paste backfills. *International Journal of Mining, Reclamation, and Environment*, 32(4), 273-296.

Reviewer: 2

The manuscript explored the characteristics of cemented backfill under different humidity curing conditions using NMR and SEM techniques, which contributes to the existing literatures. It is an interesting research that deserves to be published. However, some comments should be addressed as follows:

- 1 The introduction part needs to be extended. The Introduction should present the reasons why studying the effect of humidity in CPB is necessary, rather than just list previous publications in the related field.
- 2 The references should be updated. The authors are suggested to cite some latest literatures, as the CPB develops rapidly.
- 3 WAT2 is unclear, and it should be explained with a schematic diagram.
- 4 The sum of element contents isn't 100% In table2?
- 5 The NMR parameters are important in this manuscript and are not mentioned in the paper. The authors are suggested to provide the NMR parameters.
- 6 The authors claimed that "The distribution of the T2 value represents the pore size distribution", which is not suitable for some samples in this paper are not saturated.
- 7 Please specify the application conditions of Equation 2.

Reviewer: 3

- (1) What is the chemical composition of cement ? and it would be better to explain the derivation process of Equation.
- (2) what is the curing temperature of all samples?
- (3) which mine's UCS basic requirement is 1.4MPa, and what are the meanings of white areas and black areas in SEM binary images ?

Author's Response to Decision Letter for (RSOS-191227.R0)

See Appendix A.

RSOS-191227.R1 (Revision)

Review form: Reviewer 2

Is the manuscript scientifically sound in its present form?

Yes

Are the interpretations and conclusions justified by the results?

Yes

Is the language acceptable?

Yes

Do you have any ethical concerns with this paper?

No

Have you any concerns about statistical analyses in this paper?

No

Recommendation?

Accept as is

Comments to the Author(s)

All the comments have been addressed, and it can be plished.

Review form: Reviewer 3

Is the manuscript scientifically sound in its present form?

Yes

Are the interpretations and conclusions justified by the results?

Yes

Is the language acceptable?

Yes

Do you have any ethical concerns with this paper?

No

Have you any concerns about statistical analyses in this paper?

Yes

Recommendation?

Accept with minor revision (please list in comments)

Comments to the Author(s)

(1) please provide the original data of Fig.3, and explain that how to develop the fig.3 using the data.

(2) please add and explain that what is your research significance of the result for mine backfill application.

Decision letter (RSOS-191227.R1)

30-Oct-2019

Dear Dr Zhao:

On behalf of the Editors, I am pleased to inform you that your Manuscript RSOS-191227.R1 entitled "Microscopic characterization and strength characteristics of cemented backfill under different humidity curing conditions" has been accepted for publication in Royal Society Open Science subject to minor revision in accordance with the referee suggestions. Please find the referees' comments at the end of this email.

The reviewers and Subject Editor have recommended publication, but also suggest some minor revisions to your manuscript. Therefore, I invite you to respond to the comments and revise your manuscript.

- Ethics statement

- Data accessibility

<http://datadryad.org/submit?journalID=RSOS&manu=RSOS-191227.R1>

- Competing interests

- Authors' contributions

All submissions, other than those with a single author, must include an Authors' Contributions section which individually lists the specific contribution of each author. The list of Authors

should meet all of the following criteria; 1) substantial contributions to conception and design, or acquisition of data, or analysis and interpretation of data; 2) drafting the article or revising it critically for important intellectual content; and 3) final approval of the version to be published.

- Acknowledgements

- Funding statement

Because the schedule for publication is very tight, it is a condition of publication that you submit the revised version of your manuscript before 08-Nov-2019. Please note that the revision deadline will expire at 00.00am on this date. If you do not think you will be able to meet this date please let me know immediately.

- 1) A text file of the manuscript (tex, txt, rtf, docx or doc), references, tables (including captions) and figure captions. Do not upload a PDF as your "Main Document".
- 2) A separate electronic file of each figure (EPS or print-quality PDF preferred (either format should be produced directly from original creation package), or original software format)
- 3) Included a 100 word media summary of your paper when requested at submission. Please ensure you have entered correct contact details (email, institution and telephone) in your user account

- 4) Included the raw data to support the claims made in your paper. You can either include your data as electronic supplementary material or upload to a repository and include the relevant doi within your manuscript
- 5) All supplementary materials accompanying an accepted article will be treated as in their final form. Note that the Royal Society will neither edit nor typeset supplementary material and it will be hosted as provided. Please ensure that the supplementary material includes the paper details where possible (authors, article title, journal name).

Kind regards,
Anita Kristiansen
Editorial Coordinator
Royal Society Open Science
openscience@royalsociety.org

on behalf of R. Kerry Rowe (Subject Editor)
openscience@royalsociety.org

Associate Editor Comments to Author:

Comments to the Author:

Please respond to the remaining queries of the reviewers - in particular, you have been asked to set your work in the wider context of existing research to demonstrate more effectively how your paper moves the field forwards.

Reviewer comments to Author:

Reviewer: 3

Comments to the Author(s)

(1) please provide the original data of Fig.3, and explain that how to develop the fig.3 using the data.

(2) please add and explain that what is your research significance of the result for mine backfill application.

Reviewer: 2

Comments to the Author(s)

All the comments have been addressed, and it can be plished.

Author's Response to Decision Letter for (RSOS-191227.R1)

See Appendix B.

Decision letter (RSOS-191227.R2)

06-Nov-2019

Dear Dr Zhao,

I am pleased to inform you that your manuscript entitled "Microscopic characterization and strength characteristics of cemented backfill under different humidity curing conditions" is now accepted for publication in Royal Society Open Science.

Appendix A

Dear Editor and Reviewers:

Thank you for your letter and for the reviewers' comments concerning our manuscript entitled "Microscopic characterization and strength characteristics of cemented backfill under different humidity curing conditions" (ID: RSOS-191227). Those comments are all valuable and very helpful for revising and improving our paper, as well as the important guiding significance to our researches. We have studied comments carefully and have made correction which we hope meet with approval. Revised portion are marked in red in the paper. The main corrections in the paper and the responds to the reviewers comments are as flowing:

Responds to the reviewers comments:

(Blue indicates the problem, green indicates the original paper, and red indicates the respond.)

Reviewer #1:

Abstract

1. ...standard curing... Please explain this in more detail in this section.

"standard curing"Refers to "20°C, humidity 99%", I have given it in the revised paper.

2. What is the main outcome of this research? Please give it at the end of the abstract section.

The main outcome of this research is "It can be found that the factors affecting the strength characteristics of CPB include not only the stone powder content, but also the curing conditions of different humidity." I have given it at the end of the abstract section.

Introduction

3. Please rewrite the introduction section by focusing on problematic, suggested methods, and originality and specific objectives. Please use the present literature critically, not giving their whole details.

Thank you very much for the literature provided by the reviewer. They are very helpful to me.

Especially the following two literature:

Xue, G., Yilmaz, E., Song, W., Yilmaz, E., 2019a. Influence of fiber reinforcement on mechanical behavior and microstructural properties of cemented tailings backfill. *Constr. Build. Mat.* 213, 275-285.

Cao, S., Yilmaz, E., Song, W., 2019. Fiber type effect on strength, toughness, and microstructure of early age cemented tailings backfill. *Constr. Build. Mat.* 223, 44-54.

I have rewritten the introduction section in the revised paper.

4. At the end of this section, please provide the originality and specific objectives of this study.

The originality and specific objectives of this study are "NMR technology and an uniaxial compression test were used to explore the effects of humidity on macroscopic and microscopic characteristics and mechanical laws of CPB." And I have provided the originality and specific objectives of this study in the revised paper.

Materials and Methods

5.Please provide further information on tailings samples. Where the authors collect these samples

in the mine? They come from tailings dams or paste plant filtering cakes or mineral processing discharge products. Please state it clearly.

The experimental tailings were taken from mineral processing discharge products from the concentrator. And I have stated it clearly in the revised paper.

6. What were the sample preparation and mixing conditions? Please mention about them shortly.

The sample preparation and mixing conditions were “Various raw materials were weighed according to the designed ratio and then place them in a mixing bowl and mix well by mixer.” I have already described in the revised paper.

7. How is the manufacturing and curing process of CPB samples? Please make it simple in the revised paper.

I have simplified the manufacturing and curing process of CPB samples in the revised paper.

8. Please explain the experimental test procedures in more detail. How were the working conditions of samples? Please state them clearly in the revised paper.

the experimental test procedures has been changed to “① NMR test: An NMR analysis was conducted using MesoMR23-060H. Parameter settings were shown in Table 5. The cured specimens were tested by NMR. The NMR T2 spectrum was obtained using NMR analysis software, and the area of the T2 spectrum, the area of each peak, and its proportion were calculated. Of which the permeability parameter of samples subjected to saturation and centrifugation could be obtained by NMR.

② UCS test: The UCS test was carried out using a pressure tester No. 01000405. Its loading rate was 0.2 kN/s. And then the cured specimens were placed in the middle of the pressure plate to be tested by UCS test machine. The UCS value was obtained through a force measurement system analysis.” I have stated it clearly in the revised paper.

Table 5:NMR test parameter setting

para meter	SW/ KHz	SF/ mhz	O1/ Hz	TD	DRG1	NS	P1/ us	p2/ us	NECH	TW	RFD
Number	100	21	273173.80	2256	3	16	11.52	23.04	150	3000	0.3

9. Please add the permeability tests in this section since the authors provide some results on this test later on.

The permeability parameter of samples subjected to saturation and centrifugation could be obtained by NMR. And I have already added the permeability tests in the revised paper.

Analysis results

10. Are there any threshold limits of NMR results? If yes, please discuss it in the body of the manuscript as well as in Figure 3 (as a dotted line).

There are no threshold limits of NMR results.

11. What are the main reasons for observing this behavior in Figure 4 in terms of strength differences? Please discuss it thoroughly.

I added some discussion that when the stone powder content is the same, the UCS decreases

obviously as the increase of humidity. It shows that the moist environment seriously affects the UCS. Although the moist environment could promote the hydration reaction, at this time, the moisture content is a decisive factor affecting the strength, which results in a decrease in the strength of CPB.

Results and discussion

12. Which type of permeability tests were followed in this study? Please state it clearly.

The type of permeability tests was Coates model. And I have stated it clearly in the revised paper.

13. In Figure 5; permeability tests were conducted on fresh or hardened samples? Please clear it in the revised paper.

Permeability tests were conducted on hardened samples. And I have stated it clearly in the revised paper.

14. In Figure 7; this figure is very nice. The authors need to present in a larger size for better resolution.

I have presented in a larger size figure in the revised paper.

Conclusion

15. Please provide a short summary before highlighting the main findings of this research project.

I have provided a short summary before highlighting the main findings of this research project.

16. What are the limitations and recommendations of this work which can give some inspirations to other researchers to be worked in the area of this research?

The limitations and recommendations of this work are various measurement errors and the co-effect of temperature and humidity changes on CPB. And I have already described it in the revised paper.

Special thanks to you for your good comments.

Reviewer #2:

1. The introduction part needs to be extended. The Introduction should present the reasons why studying the effect of humidity in CPB is necessary, rather than just list previous publications in the related field.

The introduction part has been extended. I have presented the reasons why studying the effect of humidity in CPB. And I have rewritten the introduction section in the revised paper. The reasons are “The hydration of cement is affected by the level of humidity. In different stopes of underground mines, the environment humidity is not the same. Humidity affects the hydration of CPB, which affects the strength characteristics of the CPB. There are some scholars who have studied the concrete curing humidity, but there are few studies on curing humidity of CPB. And

the hydration reaction of cement in CPB under different humidity conditions, particularly the backfill in the replacement using stone powder cement, must have its own law.”

2. The references should be updated. The authors are suggested to cite some latest literatures, as the CPB develops rapidly.

I have already cited some latest literatures in the revised paper.

3. WAT2 is unclear, and it should be explained with a schematic diagram.

I have explained it with a schematic diagram in the revised paper.

4. The sum of element contents isn't 100% In table2?

There are other elements that are not listed in Table 2, so the sum of element contents isn't 100%.

5. The NMR parameters are important in this manuscript and are not mentioned in the paper. The authors are suggested to provide the NMR parameters.

I have provided the NMR parameters in the revised paper. The NMR Parameters are as follows:

Table 5:NMR test parameter setting

parameter	SW/ KHz	SF/ mhz	O1/ Hz	TD	DRG1	NS	P1/ us	p2/ us	NECH	TW	RFD
Number	100	21	273173.80	2256	3	16	11.52	23.04	150	3000	0.3

6. The authors claimed that “The distribution of the T2 value represents the pore size distribution”, which is not suitable for some samples in this paper are not saturated.

“The distribution of the T2 value represents the pore size distribution” changed to “The distribution of the T₂ value represents the pore size distribution for samples which are saturated”.

And I also changed this sentence in the revised paper.

7. Please specify the application conditions of Equation 2.

For the pores of spherical model and tube bundle model, this equation 2 can be used. And I have stated in the revised paper.

Special thanks to you for your good comments.

Reviewer #3:

1. What is the chemical composition of cement? and it would be better to explain the derivation process of Equation.

I have provided the chemical composition of cement in the revised paper. the chemical composition of cement are as follows:

Table 2: Chemical composition of cement

Chemical composition	3Ca.SiO ₂	2 Ca.SiO ₂	3CaO.Al ₂ O ₃	4CaO.Al ₂ O ₃ .Fe ₂ O ₃
Content/%	52.8	20.7	11.5	8.8

And I have described the derivation process of this Equation in the revised paper.

2. what is the curing temperature of all samples?

The curing temperature of all samples is 20°C. And I have stated in the revised paper.

3. which mine's UCS basic requirement is 1.4MPa, and what are the meanings of white areas and black areas in SEM binary images?

Gaofeng mine's UCS basic requirement is 1.4MPa. white represents pores and black represents solids in the binarization diagram. And I have stated in the revised paper. The content added in the original paper is “The binarization image can well reflect the distribution of pores. Binarization mainly uses threshold grayscale segmentation to transform the solid phase and pores in the image into black and white regions. The principle is:

$$f(i, j) = \begin{cases} 0, & f(i, j) < T \\ 1, & f(i, j) \geq T \end{cases}$$

Where: T is the threshold gray scale. By the binarization function, the pixel points whose gray value are lower than T are set to 0, that is, the black point (solid phase), and the pixel points whose gray value are higher than or equal to T are set to 1, that is, white point (porosity).”

Special thanks to you for your good comments.

Appendix B

Dear Editor and Reviewers:

Thank you for your letter and for the editor' comments and reviewer' comments concerning our manuscript entitled "Microscopic characterization and strength characteristics of cemented backfill under different humidity curing conditions" (ID: RSOS-191227). Those comments are all valuable and very helpful for revising and improving our paper, as well as the important guiding significance to our researches. We have studied comments carefully and have made correction which we hope meet with approval. Revised portion are marked in red in the paper. The main corrections in the paper and the responds to the reviewers and editor comments are as flowing:

(Red indicates the respond.)

Reviewer: 3

(1) please provide the original data of Fig.3, and explain that how to develop the fig.3 using the data.

I have provided the original data of Fig.3 in the attachment. I used the Origin software to analyze the data to get fig.3.

(2) please add and explain that what is your research significance of the result for mine backfill application.

The use of stone powder instead of cement can effectively reduce the filling cost. The research on the influence of curing humidity on CPB can provide guidance for CPB in different stopes to meet the requirements of mines. And I have already added this section in the revised paper.

Editor:

Please respond to the remaining queries of the reviewers - in particular, you have been asked to set your work in the wider context of existing research to demonstrate more effectively how your paper moves the field forwards.

I have responded to the remaining queries of the reviewers. Although there have been some researches on curing conditions of CPB, they did not study in detail the influence of curing humidity on CPB. In this paper, the influence of curing humidity on strength characteristics and microscopic characterization of CPB is studied in detail, and the existing research in this field is supplemented to make the research in this field more perfect.